# Empagliflozin Improves Metabolic and Hepatic Outcomes in a Non-Diabetic Obese Biopsy-Proven Mouse Model of Advanced NASH

**DOI:** 10.3390/ijms22126332

**Published:** 2021-06-13

**Authors:** Nikolaos Perakakis, Pavlina Chrysafi, Michael Feigh, Sanne Skovgard Veidal, Christos S. Mantzoros

**Affiliations:** 1Department of Medicine, Beth Israel Deaconess Medical Center, Harvard Medical School, 330 Brookline Ave, Boston, MA 02215, USA; nperakak@bidmc.harvard.edu (N.P.); chrysafipavlina@gmail.com (P.C.); 2Gubra, Hørsholm Kongevej 11, B, 2970 Hørsholm, Denmark; mfe@gubra.dk (M.F.); ssv@gubra.dk (S.S.V.); 3Department of Medicine, Boston VA Healthcare System, Harvard Medical School, Boston, MA 02115, USA

**Keywords:** NAFLD, steatosis, SGLT2, lipidomics, liver, NASH, diabetes

## Abstract

Empagliflozin, an established treatment for type 2 diabetes (T2DM), has shown beneficial effects on liver steatosis and fibrosis in animals and in humans with T2DM, non-alcoholic fatty liver disease (NAFLD) and steatohepatitis (NASH). However, little is known about the effects of empagliflozin on liver function in advanced NASH with liver fibrosis and without diabetes. This study aimed to assess the effects of empagliflozin on hepatic and metabolic outcomes in a diet-induced obese (DIO) and insulin-resistant but non-diabetic biopsy-confirmed mouse model of advanced NASH. Male C57BL/6JRj mice with a biopsy-confirmed steatosis and fibrosis on AMLN diet (high fat, fructose and cholesterol) for 36-weeks were randomized to receive for 12 weeks: (a) Empagliflozin (10 mg/kg/d p.o.), or (b) vehicle. Metabolic outcomes, liver pathology, markers of Kupffer and stellate cell activation and lipidomics were assessed at the treatment completion. Empagliflozin did not affect the body weight, body composition or insulin sensitivity (assessed by intraperitoneal insulin tolerance test), but significantly improved glucose homeostasis as assessed by oral glucose tolerance test in DIO-NASH mice. Empagliflozin improved modestly the NAFLD activity score compared with the vehicle, mainly by improving inflammation and without affecting steatosis, the fibrosis stage and markers of Kupffer and stellate cell activation. Empagliflozin reduced the hepatic concentrations of pro-inflammatory lactosylceramides and increased the concentrations of anti-inflammatory polyunsaturated triglycerides. Empagliflozin exerts beneficial metabolic and hepatic (mainly anti-inflammatory) effects in non-diabetic DIO-NASH mice and thus may be effective against NASH even in non-diabetic conditions.

## 1. Introduction

Non-alcoholic fatty liver disease (NAFLD) is a metabolic disease with a continuously increasing prevalence that is currently affecting approximately 30% of the general population and 40–50% of patients with type 2 diabetes (T2DM) and obesity [1,2]. Early stage NAFLD is characterized by an increased intrahepatic lipid accumulation (non-alcoholic fatty liver, NAFL) [1,2]. As the disease progresses, hepatic inflammation occurs (non-alcoholic steatohepatitis, NASH) and can lead to liver fibrosis and cirrhosis. NASH with liver fibrosis is associated with a higher liver- and cardiovascular-related mortality [1,2].

Despite the high prevalence of NASH, no specific treatment for the disease exists to date [1,2,3]. Due to the strong pathophysiological links between obesity, insulin resistance, diabetes and the development progression of NASH, many studies have focused on the investigation of already licensed anti-diabetic medications for the treatment of NASH [1,2,3,4]. Sodium-glucose cotransporter 2 inhibitors (SGLT2i) reduce blood glucose levels by inhibiting glucose reabsorption in the proximal tubule of the kidney, thus leading to glucosuria and osmotic diuresis [5]. SGLT2i are widely used for the treatment of T2DM; they lead to a moderate weight loss (2–3 kg) and they demonstrate important benefits on cardiorenal functions [5,6,7]. SGLT2i have demonstrated promising results for the treatment of NAFLD in people with T2DM [5,6,7]. Specifically, SGLT2i reduced liver fat, assessed by MRI or a controlled attenuation parameter (CAP), liver transaminases and markers of liver fibrosis (assessed by fibrosis scores or transient elastography) in patients with NAFLD and T2DM [3,5]. Among the SGLT2i, empagliflozin has been evaluated in several randomized controlled trials [8,9,10] and has been shown to reduce liver fat and liver stiffness measurements [8,9,10]. Interestingly, empagliflozin exerted these beneficial effects on the liver status in patients with T2DM even when they had excellent glycemic control and a short known disease duration [9], suggesting that its effects may be, at least partially, independent of glycemic status.

Consequently, the results from human studies raise the following questions: (a) whether empagliflozin, apart from liver fat and markers of liver fibrosis, can improve liver inflammation and fibrosis stage as assessed by the gold standard, which is liver histology; (b) whether—and if yes, through which mechanisms—empagliflozin can be an effective treatment for NAFLD in non-diabetic conditions, i.e., in pre-diabetes obesity. Animal studies with empagliflozin have been performed so far in animal models of overt diabetes [11,12,13,14]. In one case, the short-term treatment with empagliflozin was assessed in obesity with early stages of NAFLD [15]. Thus, we aimed to investigate the effects of long-term treatment with empagliflozin in a validated non-diabetic insulin-resistant obese mouse model of an histology-proven advanced NAFLD that is characterized by the presence of not only liver steatosis but also liver inflammation and fibrosis.

## 2. Results

### 2.1. Empagliflozin Has No Major Effect on Body Weight or Body Composition in DIO-NASH Mice

Both DIO-NASH mice treated with a vehicle as well as DIO-NASH mice treated with empagliflozin gained approximately 5% body weight during the 12 weeks of treatment whereas mice fed a chow diet gained significantly less weight (Figure 1). The higher body weight derived from an increase exclusively in fat mass both in the vehicle- and in the empagliflozin-treated group induced by the AMLN diet, with no changes in lean mass or water mass (Figure 1).

### 2.2. Empagliflozin Improves Glucose Homeostasis Without Affecting Insulin Sensitivity in DIO-NASH Mice

DIO-NASH mice treated with a vehicle had slightly higher fasting glucose levels at the treatment start (week 0) and a similar response in the oral glucose tolerance test (OGTT) at week 7 compared with the chow mice thus suggesting the presence of a mild hyperglycemia but no overt diabetes (Figure 2). Despite the presence only of a mild hyperglycemia, treatment with empagliflozin resulted in lower glucose levels at 0, 15, 30, 120 and 180 min in OGTT at week 7 and lower fasting glucose levels at week 12 compared with the vehicle (Figure 2). The DIO-NASH mice treated with the vehicle were insulin-resistant, indicated by the higher fasting insulin levels, higher homeostatic model assessment of insulin resistance (HOMA-IR) and lower quantitative insulin sensitivity check index (QUICKI) score both at the treatment start (week 0) and the treatment completion (week 12) as well as by the lower calculated plasma glucose disappearance rate (KITT) in the ipITT at week 10 compared with the vehicle (Figure 3). Treatment with empagliflozin led to lower glucose levels at 0, 15, 30, 60, 120 and 180min of the ipITT test at week 10 compared with the vehicle but it did not improve KITT in the ipITT at week 10 or HOMA-IR and the QUICKI score at the treatment start (week 0) and the completion (week 12) (Figure 3). These findings suggest that empagliflozin reduced the blood glucose levels but did not affect insulin sensitivity. Empagliflozin had no effect on the concentrations of liver transaminases and on the blood lipid profile (Appendix A).

### 2.3. Empagliflozin Improves Histopathological NAFLD Activity Score (NAS) But Not Fibrosis Stage In DIO-NASH Mice

All DIO-NASH mice included in both groups in our study had a NAS ≥ 4 (in most cases 5 or 6) and liver fibrosis. Treatment with empagliflozin resulted in a significant reduction in NAS compared with treatment with the vehicle in the post- vs. pre-treatment biopsy (Figure 4). Interestingly, this reduction was achieved by an improvement in the lobular inflammation score (Figure 4). No change in the steatosis score in the post- vs. pre-treatment biopsy was observed in the DIO-NASH mice treated with empagliflozin or the vehicle. In agreement with the assessment of liver steatosis in hematoxylin-eosin (H&E) staining, there were no differences in liver weight and in the hepatic lipid content and total cholesterol (Appendix A). Treatment with empagliflozin in the DIO-NASH mice had no effect on the fibrosis stage (Figure 2A and Appendix A). Additionally, empagliflozin did not affect the markers of Kupffer and stellate cell functions such as Galectin-3, alpha-smooth muscle actin (a-SMA), collagen 1a1 (Col1a1) and hydroxyproline, suggesting that these cells had a minor contribution to the anti-inflammatory effects of empagliflozin (Appendix A).

### 2.4. Empagliflozin Reduces Lactosylceramides and Increases Unsaturated Triglycerides

Although empagliflozin did not significantly affect the lipid content, we performed a hepatic lipidomic analysis in order to assess whether empagliflozin affected the lipid composition. The lipidomic analysis of the liver identified 987 lipid species. The partial least squares-discriminant analysis (PLS-DA) showed that, based on the lipid measurements, the DIO-NASH + empagliflozin group and the DIO-NASH + vehicle group formed two distinct clusters with some overlap in their 95% confidence interval (Figure 5). This suggested the presence of several relevant differences in the hepatic lipid profile between the two groups. In a further analysis, we identified that DIO-NASH mice treated with empagliflozin had lower concentrations of hepatic lactosylceramides and specifically of the species LCER (16:0) and LCER (18:0). Additionally, we observed significantly higher concentrations of triglyceride species that contained one or more unsaturated fatty acids (Figure 5).

## 3. Discussion

We demonstrate herein that treatment with empagliflozin improved glucose metabolism and liver pathology in biopsy-confirmed DIO-NASH mice with insulin resistance without the presence of diabetes.

Previous studies in different animal models of diabetes with obesity (e.g., streptozotocin-induced [11,16], db/db [12], ApoE^(−/−)^ mice [13] and OLETF rats [14]) have shown that treatment with empagliflozin profoundly improves liver steatosis and inflammation by enhancing macrophage autophagy and macrophage polarization to the anti-inflammatory M2 phenotype [12,13,17] as well as by reducing the expression of genes involved in endoplasmic reticulum stress [13,18], lipogenesis [13,14,18] and gluconeogenesis [14]. In these animal models of diabetes with obesity, liver fibrosis was not assessed whereas treatment with empagliflozin profoundly reduced both the severe high (in most cases above 200–250 mg/dL) glucose levels as well as the body weight. Thus, the beneficial effects of empagliflozin on the liver pathology in these animal models resulted most probably from the observed robust improvements in glucose and energy homeostasis.

In our study, we used a model that developed diet-induced obesity, insulin resistance and NASH with steatosis, inflammation and liver fibrosis but no overt diabetes. Despite the fact that treatment with empagliflozin did not reduce body weight and improved only slightly blood glucose levels, it still led to a modest but significant reduction in NAS in the liver pathology compared with the vehicle. Interestingly, the reduction in NAS by empagliflozin was driven by lower scores in liver lobular inflammation and not in liver steatosis. Treatment with empagliflozin had, in these conditions, no effect on liver fibrosis. To our knowledge, only one study has previously assessed the effects of empagliflozin alone or combined with dulaglutide in a non-diabetic mouse model of NAFLD with inflammation (NASH) and did not find any major impact of empagliflozin on liver histology [15]. However, in that study, the mice had a milder NAFLD phenotype (NAS score 3–4) and were treated for a much shorter period of time compared with our study (4 weeks vs. 12 weeks). Finally, the liver pathology was evaluated only at the treatment completion and not additionally at the treatment initiation (pre-to-post), as in our study. Nevertheless, the reported effects of both studies argue for mild effects of empagliflozin in non-diabetic conditions, which thus necessitates a longer duration of treatment in order to achieve relevant hepatic benefits. In agreement with the above, when comparing the effects of empagliflozin with the effects of medications of other classes (i.e., liraglutide (glucagon-like peptide 1 receptor analogue), elafibranor (peroxisome proliferator-activated receptor alpha/delta agonist) or CHS-131 (selective peroxisome proliferator-activated receptor gamma modulator)) in the same mouse model experimental setting and for the same duration of treatment [19,20], the improvement in NAS score was more modest with empagliflozin, which suggested again weaker hepatic actions at normal/slightly elevated glucose levels compared with the other medications.

When assessing the possible mechanisms mediating the effects of empagliflozin, we did not observe any changes in the markers of the hepatic Kupffer (hepatic macrophages) and stellate cell activation as well as in the amount of intrahepatic lipid content. We did, however, observe changes in the intrahepatic lipid composition with empagliflozin, which were characterized by a reduction in the hepatic concentrations of lactosylceramides and an increase in the hepatic concentrations of several unsaturated triglyceride species. Hepatic lactosylceramides are elevated in NASH in humans [21] and are considered important contributors to inflammation by regulating cell adhesion/migration pathways [22]. Among their different pro-inflammatory functions, lactosylceramides can disrupt autophagy procedures [22,23]. Thus, the reduction in their levels with empagliflozin in our study might contribute to the improvement in hepatic inflammation, possibly through the enhancement of autophagy as reported previously in the diabetic mouse models of NAFLD [12,13]. Additionally, the increase in the concentrations of intrahepatic triglyceride species with unsaturated fatty acids might also have an anti-inflammatory role. Specifically, the metabolism of polyunsaturated fatty acids can lead to the formation of specialized pro-resolving mediators (SPMs) that have multiple anti-inflammatory properties [24]. Future studies should address whether the changes we observed in lactosylceramides and unsaturated triglyceride species with empagliflozin are the main triggers of enhanced autophagy and increased synthesis of SPMs with the medication (Figure 6).

Although the hepatic effects of empagliflozin are more modest compared with other medications under the same conditions, they are mediated through distinct mechanisms, which supports the notion for the future evaluation of empagliflozin as part of a combination therapy. Furthermore, our findings supported the evaluation of empagliflozin in humans with NAFLD, obesity-insulin resistance and pre-diabetes who may profit from the combined beneficial effects of the medication in the cardiovascular, renal and metabolic system as well as in the hepatic function.

Our study has several strengths and limitations. A strength of the study is the use of an established mouse model that develops all of the components of human NAFLD. Another strength is the use of a liver biopsy both before and after treatment completion, which allowed us to include our study mice with advanced NAFLD and to perform comparative evaluations. A limitation of our study is the lack of direct assessment of liver autophagy or SPM levels.

In conclusion, we have shown that empagliflozin exerts beneficial metabolic and hepatic (mainly anti-inflammatory) effects in non-diabetic mice with diet-induced obesity, insulin resistance and NASH, possibly by downregulating the intrahepatic concentrations of lactosylceramides and by increasing hepatic unsaturated triglycerides (Figure 4). Future studies should assess whether empagliflozin can be an effective treatment, alone or in combination, in humans with NAFLD and insulin resistance pre-diabetes.

## 4. Methods

### 4.1. Study Design

The study design and methodology have been previously described [19,20]. Briefly, male C57BL/6JRj mice were fed an AMLN diet (D09100301, Research Diets Inc., New Brunswick, NJ, USA) that consisted of 40% fat, 2% cholesterol and 22% fructose for 36 weeks before the study initiation. At −3 weeks, a pre-biopsy was performed. The mice were anesthetized with isoflurane (2–3%) in 100% oxygen, a small abdominal incision was made in the midline and a cone shaped wedge of liver tissue (approximately 50 mg) was excised from the distal portion of the left lateral liver lobe, which was then fixated in 10% neutral buffered formalin (4% formaldehyde). Only mice with a steatosis score ≥ 2 and a fibrosis score ≥ 1 according to criteria by Kleiner et al. [25] were included and randomized to receive the vehicle or different treatments for 12 weeks. Another group that consisted of mice of a similar age fed a normal chow diet was also included. The results regarding the other treatments have been previously published [19,20]. In the current analysis, we focused on DIO-NASH mice that received empagliflozin (*n* = 13, DIO-NASH + empagliflozin) at a dose of 10 mg/kg/day and we compared the findings with DIO-NASH mice treated with the vehicle (*n* = 13, DIO-NASH + vehicle) for 12 weeks. Finally, in order to compare the metabolic status of the DIO-NASH mice, we included in our analysis a group of mice fed a normal chow diet (*n* = 12, Chow + vehicle). All mouse experiments were performed according to Gubra bioethical guidelines that fully comply with international accepted principles for the care and use of laboratory animals.

### 4.2. Biochemical Measurements and Tolerance Tests

The measurements and procedures are described with detail in the supplemental information in [19]. Briefly, plasma and hepatic (after homogenization and extraction) parameters, i.e., alanine aminotransferase (ALT), aspartate aminotransferase (AST), total cholesterol and triglycerides, were measured with an automated analyzer (Cobas c501, Roche Diagnostics, Basel, Switzerland), insulin with the MSD Platform (Meso Scale Diagnostics, Rockville, MD, USA) and hydroxyproline with a colorimetric assay (QuickZyme Biosciences, Leiden, The Netherlands). The OGTT and the ipITT were performed after a 6 h fast at week 7 and week 10 of treatment, respectively.

### 4.3. Liver Pathology and Immunohistochemistry (IHC) Staining

Details about the liver pathology and IHC staining have been described in [19,20]. Briefly, mice were terminated at 12 weeks and, upon necropsy, the whole liver was collected, weighed and sampled for further analyses. The liver post-biopsy for histological analyses was obtained from the left lateral lobe, fixated in 4% formalin for 20–24h and embedded in paraffin. The liver biopsies for liver triglycerides and total cholesterol were dissected from the medial lobe and the liver biopsies for hydroxyproline were dissected from the entire caudal lobe, snap frozen in liquid nitrogen and stored at −80 °C. The liver biopsies (before treatment initiation and at treatment completion) based on H&E and Picrosirius red staining were scored according to Kleiner et al. [25] by an experienced histopathologist who was blinded for the treatment groups. A quantitative assessment of steatosis in H&E, of Galectin-3, a-SMA and Col1a1 in IHC was performed with Visiomorph software as described in the supplemental appendix in [19].

### 4.4. EchoMRI Body Composition

The body composition was assessed by an EchoMRI 3-1 body composition analyzer (EchoMRI, Huston, TX, USA) in non-anaesthetized mice.

### 4.5. Lipidomics

A mass-spectrometric analysis of hepatic lipids (lipidomics) was performed by Metabolon Inc. (Morrisville, NC, USA) as described in detail in [19,20].

### 4.6. Statistics

A statistical analysis was performed with GraphPad Prism 8 (GraphPad Software Inc., La Jolla, CA, USA) and MetaboAnalystR [26]. With *n* = 13 for DIO-NASH + vehicle and DIO-NASH + empagliflozin, the study had 80% power to detect differences ≥ 20% of the mean with an SD of 20% of the mean and at an α = 0.05 two-tailed. A one- or two-way (for OGTT and ipITT) ANOVA and mixed-models (due to missing values) for changes in body weight were performed with parameters Time (corresponding with days for body weight and with minutes for OGTT and ipITT) and Treatment (DIO-NASH + empagliflozin, DIO-NASH + vehicle, Chow + vehicle). Post-hoc Dunnett tests for the comparison of DIO-NASH + empagliflozin vs. DIO-NASH + vehicle or for Chow + vehicle vs. DIO-NASH + vehicle were performed when three groups were involved and when *p* by the interaction of Time with Treatment (Time*Treatment) was <0.05. An unpaired *t*-test was performed for the parameters where only two groups (DIO-NASH + empagliflozin, DIO-NASH + vehicle) were assessed and after the confirmation of the normal distribution for the involved parameters with a D’Agostino–Pearson test. One-tailed *p*-values in the unpaired *t*-test were reported for the histologic outcomes of DIO-NASH + empagliflozin (Figure 4) based on the a priori hypothesis that these effects would be one-directional, i.e., beneficial and not detrimental. The a priori hypothesis was based on the fact that no detrimental effects of empagliflozin on the liver function have been reported so far: (a) in numerous animal studies (in models with and without NAFLD); (b) in human Phase II and Phase III randomized clinical trials (RCTs) with patients with NAFLD and T2DM including a study with patients that had excellent glycemic control and short known disease duration [9]; (c) in Phase III RCTs that evaluated the effects of empagliflozin on renal and cardiovascular outcomes in non-diabetic patients. Two-tailed *p*-values were reported for all other parameters including lipidomics. Regarding the lipidomics, we performed the analysis as previously described [19,20,27,28]. The analysis included data curation, PLS-DA and a univariate analysis. Regarding the triglyceride species in Figure 2B, only the ones with a fold increase above 1.3 for treatment with empagliflozin compared with the vehicle were demonstrated. The QUICKI score was calculated based on the equation QUICKI = 1/[log(FI) + log(FG)] where FI was the fasting insulin expressed in μU/mL and FG was the fasting glucose expressed in mg/dl. The glucose decline rate for ITT (KITT %/min) was calculated as KITT = (0.693 × 100)/t1/2, where t1/2 was the glucose half-life. In order to calculate t1/2, a simple linear regression of ITT for up to 60 min (timepoint that glucose levels reached a plateau and the 50% reduction of glucose had been completed) was performed for each group (i.e., chow + vehicle, DIO-NASH + vehicle, DIO-NASH + empagliflozin) in GraphPad Prism 8.

## 5. Conclusions

The present study demonstrates the beneficial metabolic and hepatic (mainly anti-inflammatory) effects of empagliflozin in non-diabetic obese mice with insulin resistance and NASH. Further research is required to elucidate the underlying mechanisms and to evaluate whether empagliflozin could be effective in the treatment of NAFLD/NASH in humans even without diabetes.

## Figures and Tables

**Figure 1 ijms-22-06332-f001:**
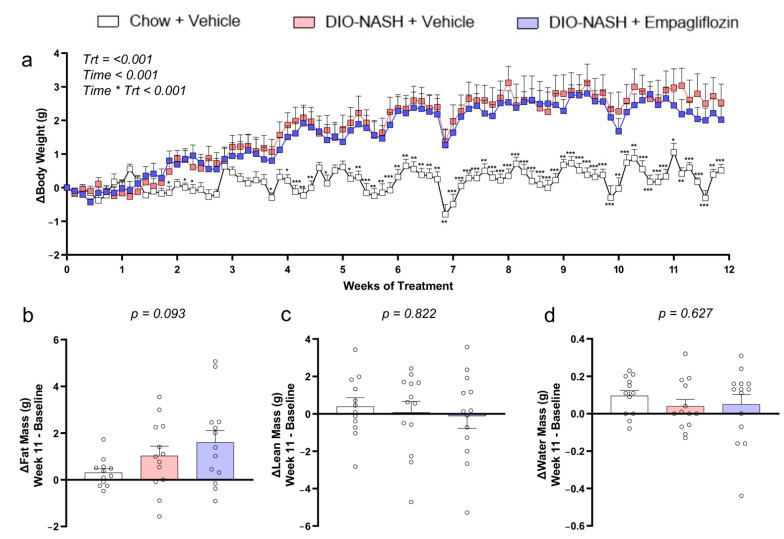
Empagliflozin does not affect the body weight and body composition in DIO-NASH mice. (**a**) Changes in body weight, (**b**) fat mass, (**c**) lean mass and (**d**) water mass during the 12 weeks of treatment. A mixed-model analysis was performed for the changes in body weight and *p*-values are reported for Trt, treatment (Chow + vehicle, DIO-NASH + vehicle, DIO-NASH + empagliflozin), Time (day of treatment) and their interaction Time * Trt. A one-way ANOVA was performed for (**b**–**d**) and *p* of the ANOVA is reported. Data show means ± SEMs. Δ refers to a change at that timepoint compared with the baseline (treatment start). Ν = 12 for Chow + vehicle and *n* = 13 for DIO-NASH + vehicle and DIO-NASH + empagliflozin.

**Figure 2 ijms-22-06332-f002:**
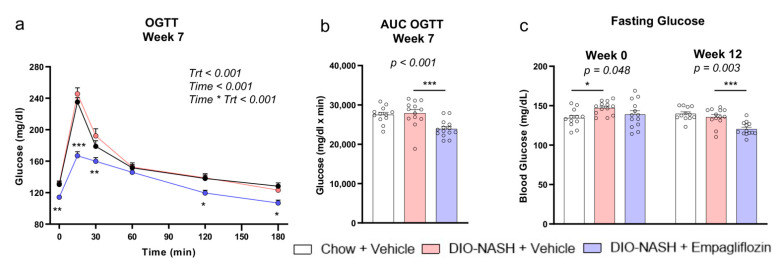
Empagliflozin improves glucose levels in DIO-NASH mice. (**a**) Glucose levels and (**b**) AUC for glucose in OGTT at week 7 and (**c**) glucose levels after 4 h fasting at week 0 (start of treatment) and week 12 (completion of treatment). A two-way ANOVA was performed for OGTT and *p*-values are reported for Trt, treatment (Chow + vehicle, DIO-NASH + vehicle, DIO-NASH + empagliflozin), Time (minutes of OGTT) and their interaction Time * Trt. A one-way ANOVA was performed for the AUC and the fasting glucose and *p* of the ANOVA is reported. By *p* < 0.05 in Trt * Time for OGTT and in the ANOVA for the AUC and for the fasting glucose, post-hoc Dunnett tests were performed in each timepoint in OGTT, at week 7 in the AUC OGTT and at week 0 and week 12 in the fasting glucose and * *p* < 0.05, ** *p* < 0.01, *** *p* < 0.001, respectively, for Chow + vehicle or DIO-NASH + empagliflozin compared with DIO-NASH + vehicle. Data show means ± SEMs. Ν = 12 for Chow + vehicle and *n* = 13 for DIO-NASH + vehicle and DIO-NASH + empagliflozin.

**Figure 3 ijms-22-06332-f003:**
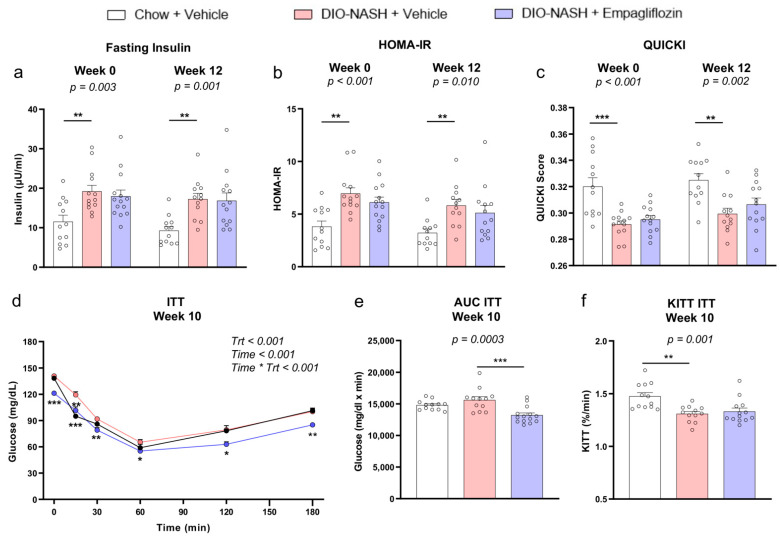
Empagliflozin does not improve insulin sensitivity in DIO-NASH insulin-resistant mice. (**a**) Insulin levels after 4 h fasting, (**b**) HOMA-IR and (**c**) QUICKI score at week 0 (start of treatment) and week 12 (completion of treatment). (**d**) Glucose levels, (**e**) AUC and (**f**) KITT in the ipITT at week 10. A two-way ANOVA was performed for the ipITT and *p*-values are reported for Trt, treatment (Chow + vehicle, DIO-NASH + vehicle, DIO-NASH + empagliflozin), Time (minutes of ITT) and their interaction Time * Trt. A one-way ANOVA was performed for all of the other parameters (**a**–**c**,**e**,**f**). The AUC and *p* of the ANOVA are reported. By *p* < 0.05 in Trt * Time for the ipITT and in the ANOVA for the other parameters, post-hoc Dunnett tests were performed in each timepoint in the ipITT, at week 10 in the AUC ITT and KITT ITT and at week 0 and week 12 in the fasting insulin, HOMA-IR and QUICKI and * *p* < 0.05, ** *p* < 0.01, *** *p* < 0.001, respectively, for Chow + vehicle or DIO-NASH + empagliflozin compared with DIO-NASH + vehicle. Data show means ± SEMs. AUC, area under the curve. Ν = 12 for Chow + vehicle and *n* = 13 for DIO-NASH + vehicle and DIO-NASH + empagliflozin.

**Figure 4 ijms-22-06332-f004:**
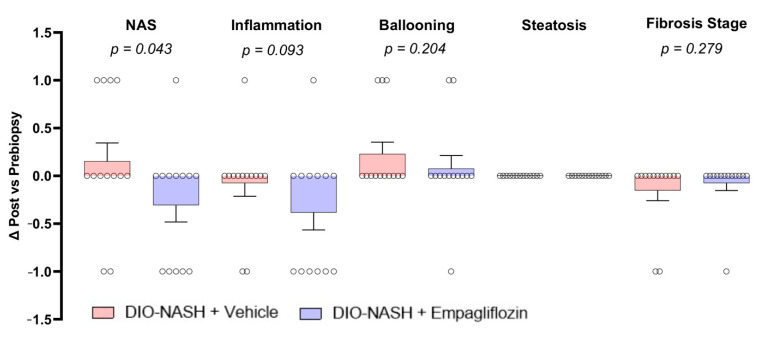
Empagliflozin improves liver histopathology. Histological changes in NAS, lobular inflammation, ballooning degeneration, steatosis and fibrosis stage in liver post- vs. pre-treatment biopsies. H&E staining for DIO-NASH mice treated with the vehicle or empagliflozin. Red arrows show inflammatory foci. Normal distribution was evaluated with a D’Agostino–Pearson test and an unpaired *t*-test was performed and one-tailed *p*-values are reported. Data show means ± SEMs. *n* = 13 for DIO-NASH + vehicle and DIO-NASH + empagliflozin.

**Figure 5 ijms-22-06332-f005:**
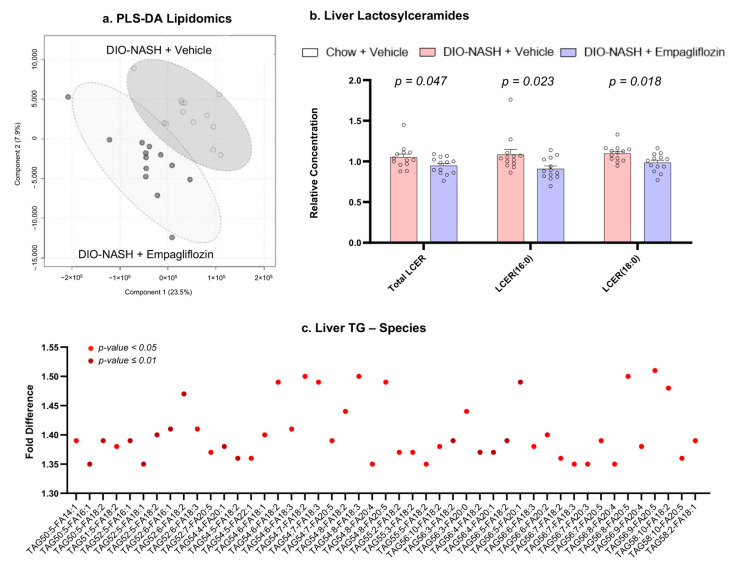
Empagliflozin affects hepatic lipidomes. (**a**) Score plot of the partial least squares-discriminant analysis (PLS-DA) showing the formation of two distinct clusters based on the treatment. Each black dot corresponds with a mouse treated with empagliflozin and each white dot with a mouse treated with the vehicle. The circles correspond with a 95% confidence interval for each group. (**b**) Relative concentrations of lactosylceramides in the liver in DIO-NASH mice treated with the vehicle and empagliflozin. (**c**) Fold difference of the concentrations of triglyceride species in DIO-NASH mice treated with empagliflozin compared with the vehicle. Triglyceride species with a fold difference > 1.3 are demonstrated (light red for *p* in *t*-test < 0.05 and dark red for *p* < 0.01). In triglyceride species, red dots correspond with *p* < 0.05 and dark red dots with *p* ≤ 0.01. *n* = 13 for DIO-NASH + vehicle and DIO-NASH + empagliflozin.

**Figure 6 ijms-22-06332-f006:**
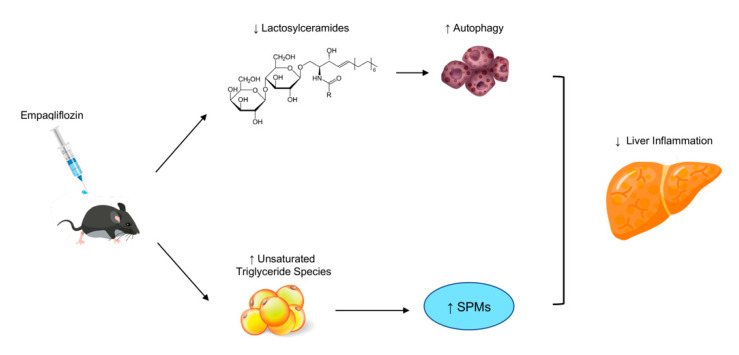
Potential mechanism of the function for empagliflozin in NASH. Lactosylceramides act as pro-inflammatory mediators and disruptors of autophagy procedures [22,23]. Treatment with empagliflozin in our study decreased the hepatic concentrations of lactosylceramides, which may explain the enhancement of autophagy that has been observed in previous diabetic mouse models of NAFLD treated with empagliflozin [12,13]. Additionally, treatment with empagliflozin increased the concentrations of triglyceride species consisting of unsaturated fatty acids. The metabolism of unsaturated fatty acids has been associated with the formation of specialized pro-resolving mediators (SPMs) that demonstrate anti-inflammatory properties [24].

## Data Availability

Data available upon request.

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
