# Peer review of "Empagliflozin Improves Metabolic and Hepatic Outcomes in a Non-Diabetic Obese Biopsy-Proven Mouse Model of Advanced NASH"

_ijms, 2021, doi:10.3390/ijms22126332_

Round 1
Reviewer 1 Report
NAFLD is one of the three main causes of cirrhosis and an increasing problem in our society. It is important to understand the links between the regulation of autophagy and the hepatic complications of NAFLD. The authors designed a study to assess the use of Empagliflozin in an obese, insulin resistant, non-diabetic mouse model of NASH. The study showed improvement in inflammation and insulin sensitivity without affecting steatosis or fibrosis. There was no affect on body weight, but NAS scores improved. I suspect that the insulin sensitivity improvement with lower scores in lobular inflammation would have improved the markers of mitochondrial function that would have accompanied this. It would be interesting to see what this effect might have been on a dose response curve as well as combo therapy in the future. I think this deserves to be published.
Author Response
Thank you very much!
Reviewer 2 Report
Empagliflozin, a sodium-glucose cotransporter-2 (SGLT2) inhibitor, has been used to reduce blood glucose by inducing renal glycosuria. The authors aim to explore whether treatment with empagliflozin could improve glucose metabolism, insulin sensitivity and liver pathology in biopsy-confirmed DIO-NASH mice with insulin resistance without the presence of diabetes. In generally, authors provided a lot of data in this study and try to demonstrate the beneficial use of empagliflozin in NASH; however, I still have significant concern about if it was carrying an aspect of novelty after major revisions.
Comments:
- Sample size need to be shown in all the figures. Moreover, how do the authors determine the sample size, it should has a scientific quantification. Every individual data/mouse should be shown in bar graph as dot plot to display the distribution of the samples.
- All the figures need to be rearranged. The figures displayed different quantitative meaning should not be arranged in the same category.
- In the line graph of Figure 1A, only one example was showed for each group. The mean value with SD or SEM should be display instead of one example, and Two-way ANOVA can be use to identify the difference between the groups.
- Authors claim “treatment with empagliflozin led to lower glucose levels at 0, 15, 30, 60, 120 and 180min of ipITT test at week 107 10 compared to vehicle which suggests improved insulin sensitivity”, however, lower glycemia was detected at 0 time point (self-control), I wonder whether empagliflozin can enhance insulin sensitivity. Because empagliflozin has caused lower glycemia, authors should use another assay to examine insulin sensitivity such as Rapid Insulin Sensitivity Test (RIST), insulin-suppression test (IST), and continuous infusion of glucose with model assessment (CIGMA).
- Additionally, only the significant difference at individual time point was shown in Figure 1C (ITT graph), however, does it have statistic significance between these three curves? Authors should display the data and explain it. The slope of the linear decline in plasma glucose (KITT) and the plasma glucose half-time (50% from baseline) should be calculated. HOMA index was used for insulin sensitivity, how about other indexes, such as QUICKI, Matsuda and DeFronzo, and Cederholm and Wibell index?
- In supplymentary Fig 1., authors need to add control group with normal chow and vehicle treatment, or it will be difficult to know the physiological change under empaglifiozin.
- In Figure 2A, not only the quantitative data, but also all the histological image data should be included in manuscript.
- Is there any change about TG-species in blood? This is a critical data to compare the liver TG-profile, the authors need to show it.
- t-test was performed for the parameters with only two groups in Figure 2. Do the samples show normal distribution? Can t-test be conducted in Figure 2? The relative information need to show in the figure legends or main text.
- The authors claimed that “One‐tailed p values in the unpaired t-test are reported for the histologic outcomes of DIO-286 NASH + empagliflozin (Figure 2A) based on the a priori hypothesis that these effects will be one‐directional, i.e., beneficial and not detrimental.”. However, DIO-NASH mice with insulin resistance without the presence of diabetes mouse model is the first time to be used for examination the function of empagliflozin, which means that priori hypothesis can not be applied in this study, and two-tail unpaired t-test should be used here.
- Based on the property of IJMS, the authors need to identify a molecule pathway to explain why empaglifiozin can improve NASH in their mouse model.
Round 2
Reviewer 2 Report
Authors has answered my requirements, I have no more questions.